# Compliance between Food and Feed Safety: Eight-Year Survey (2013–2021) of Aflatoxin M1 in Raw Milk and Aflatoxin B1 in Feed in Northern Italy

**DOI:** 10.3390/toxins15030168

**Published:** 2023-02-21

**Authors:** Luca Ferrari, Nicoletta Rizzi, Elisa Grandi, Eleonora Clerici, Erica Tirloni, Simone Stella, Cristian Edoardo Maria Bernardi, Luciano Pinotti

**Affiliations:** 1Department of Veterinary Medicine and Animal Sciences (DIVAS), Università degli Studi di Milano, 26900 Lodi, Italy; 2Associazione Regionale Allevatori della Lombardia (ARAL), 26013 Crema, Italy; 3CRC I-WE (Coordinating Research Centre: Innovation for Well-Being and Environment), Università degli Studi di Milano, 20133 Milan, Italy

**Keywords:** aflatoxins, AFM1, mycotoxins, monitoring program

## Abstract

Aflatoxins (AFs) are fungal metabolites that are found in feed and food. When ruminants eat feed contaminated with aflatoxin B1 (AFB1), it is metabolised and aflatoxin M1 (AFM1) is excreted in the milk. Aflatoxins can result in hepatotoxic, carcinogenic, and immunosuppressive effects. The European Union thus set a low threshold limit (50 ng/L) for presence of AFM1 in milk. This was in view of its possible presence also in dairy products and that quantification of these toxins is mandatory for milk suppliers. In the present study, a total of 95,882 samples of whole raw milk, collected in northern Italy between 2013 and 2021, were evaluated for presence of AFM1 using an ELISA (enzyme-linked immunosorbent assay) method. The study also evaluated the relationship between feed materials collected from the same farms in the same area during the same period (2013–2021) and milk contamination. Only 667 milk samples out of 95,882 samples analysed (0.7%) showed AFM1 values higher than the EU threshold limit of 50 ng/L. A total of 390 samples (0.4%) showed values between 40 and 50 ng/L, thus requiring corrective action despite not surpassing the regulatory threshold. Combining feed contamination and milk contamination data, some feedingstuffs seem to be more effective in defying potential carryover of AFs from feed to milk. Combining the results, it can be concluded that a robust monitoring system that covers both feed, with a special focus on high risk/sentinel matrices, and milk is essential to guarantee high quality and safety standards of dairy products.

## 1. Introduction

Mycotoxins are secondary metabolites typically produced by mould or filamentous fungi [1,2], mainly by *Aspergillus*, *Penicillium*, and *Fusarium* spp. [3]. Mycotoxins are produced following sudden changes in humidity, temperature (e.g., between day and night), or mechanical breakage of the mould itself [4]. Mycotoxins have a variety of possible adverse effects on both human and animal health [5,6].

Aflatoxins (AFs) are the most abundant group of natural fungal toxins [7]. AFs exert hepatotoxic, carcinogenic, and immunosuppressive effects and are mainly produced by *Aspergillus flavus*, *Aspergillus parasiticus*, and rarely by *Aspergillus nominus* [8,9]. These fungi are ubiquitous in nature and are found in a wide range of agricultural commodities, such as corn, wheat, corn gluten, and cotton seeds [10]. Aflatoxin B1 (AFB1) is the most dangerous mycotoxin with the highest degree of toxicity for animals. The International Agency for Research on Cancer (IARC) classified AFB1 in Group 1 as a human carcinogen [11]. However, several metabolites originate from metabolism of AFB1, of which aflatoxin M1 (AFM1) is the main product.

Cow’s milk is a nutritious biological fluid, being rich in high-quality proteins, lipids, minerals, (calcium, magnesium, selenium), and vitamins (riboflavin, vitamin B12, pantothenic acid) [12,13]. Consequently, milk is a nutrient-dense foodstuff that (along with dairy products) represents a basic food for human nutrition and development [12,14]. In addition to nutrition, milk provides several biological attributes, such as antimicrobial properties, immune stimulants, enzymes, and antibodies [13]. Presence of AFM1 in milk and dairy products, therefore, poses a potential threat to human health, especially infants and children, as they are the major consumers of milk and dairy products [15]. Although the toxicity of AFM1 is lower than that of the parent compound (AFB1) [16], AFM1 has been defined as a probable human hepatocarcinogen and is classified in Group 2B by the IARC [11]. Since milk products are consumed daily, several food safety authorities worldwide have established maximum residue limits (MRLs) for presence of AFM1 in milk and other dairy products [17]. For example, the US Food and Drug Administration (USFDA) and China have set the AFM1 action level in milk at 500 ng/L [18], whereas the European Union (EU) opted for a stricter regimen to protect consumers’ health by setting the maximum at 50 ng/L in milk [19]. Strict legal limits, such as those imposed by the EU, cause AFM1 to have adverse effects on the economy. For instance, Serbia had an AFM1 outbreak in 2013 that resulted in product recalls and a dramatic reduction in purchases of milk and dairy products. Popovic et al. (2017) estimated a total loss of between EUR 74.7 and EUR 96.2 million during this two-year crisis [20].

Development of reliable methods for determination and quantification of AFM1 in milk and dairy products has, therefore, become critical. High-performance liquid chromatography (HPLC) can be coupled with ultraviolet (UV) or fluorescence (FL) detectors. These methods are not optimal for screening several samples compared with immunological-based methods, such as enzyme-linked immunosorbent assay (ELISA) [21,22,23], immunochromatographic assay (ICA) [24,25,26], or lateral flow immunoassay (LFIA) [27,28]. The need to analyse several samples, providing fast and reliable results, as well as the need to use the lowest possible volume of sample, make ELISA the best method. In fact, although both ELISA and HPLC were shown to be suitable methods for mycotoxin analysis, the choice of one of these methods should primarily be determined by number of samples [29]. ELISA is an established high-throughput assay with low sample volume requirements and often has fewer sample clean-up procedures than the HPLC methods.

Despite efforts to control fungal contamination in feed commodities, AFs remain the most dangerous recurring mycotoxins [4]. Several feed ingredients can be contaminated with AFs, especially in countries characterised by tropical and subtropical climates. However, climate change is changing the paradigm and, in the last decade, Italy and other EU countries also have temperature and moisture conditions that support production of AFB1 by toxigenic *Aspergillus* spp. [15]. Feedstuff is thus at high risk of AFB1 contamination, which may increase the risk of AFM1 occurrence in milk.

In the current study, we report the results of the AFM1 monitoring program conducted in Lombardy Region on several raw milk samples by the Regional Breeders Association of Lombardy on cow’s milk during an eight-year period, from 2013 to 2021. Furthermore, we wanted to determine whether the differences in AFM1 levels in milk complied with presence of AFB1 in feed ingredients in the same geographical area.

## 2. Results

The presence of AFM1 in milk samples is shown in Table 1. The milk samples analysed were divided into four groups depending on level of safety/contamination: (i) very low risk (<25 ng/L), (ii) low risk (25–40 ng/L), (iii) moderate risk (40–50 ng/L), (iv) high risk (>50 ng/L).

Only 667 milk samples out of 95,882 samples analysed (corresponding to 0.7% of the total number of samples analysed) showed a content of AFM1 exceeding the EU threshold of 50 ng/L, whereas all the other samples (99.3%) complied with EU legislation. However, 390 of these samples (0.4% of the total number of samples analysed) showed AFM1 contamination between 40 and 50 ng/L.

Looking at the mean values of AFM1 contamination found over the years, milk showed the highest concentration of AFM1 in 2013 and 2015, when the mean contamination reached 14.4 and 15.1 ng/L, respectively (Figure 1).

In Figure 1, AFs measured in milk (AFM1) have been plotted with AFB1 in feed materials and forages. The results indicate that, in general there are some similarities in the pattern of AFs in both feed and milk, with some differences. Indeed, in the case of maize and cotton, the distribution level of contamination showed a parallel pattern between the test material (milk and feedingstuffs); the same did not occur in the case of the other tested feed materials, such as maize silage. Of note is the situation recorded in 2015, when an increase in AFM1 in milk was observed in conjunction with a “peak” in AFB1 contamination in cotton and maize in the form of flour and grain (Figure 1). On the other hand, silage maize was less contaminated in comparison with values recoded for AFM1 in milk (Figure 1).

Focusing on levels of AFM1 in milk from 2016 onwards, the year the regional surveillance plan was introduced, the percentage of positive samples and samples under alert (between 40–50 ng/L) is consistent over the years, while, in 2013 and 2015, there was a “peak” in AFM1 contamination in milk samples considered at moderate (40–50 ng/L) and high risk (>50 ng/L).

Combining the results of mean AFM1 contamination and percentage of positive samples over the years suggests that monitoring control programs, such as that of the Regional Breeders Association of Lombardy, help to maintain contamination of cow’s milk under safety levels for human consumption.

## 3. Discussion

Milk is a highly nutritious food, with great health benefits for human infants and children [14]. Milk is a valuable source of many macro- and micronutrients, such as minerals, fats, amino acids, and vitamins, which help individuals meet their recommended daily intake of essential nutrients for growth and maintenance of human health [30,31,32]. Wholesomeness of milk as well as its safety are, therefore, of utmost importance.

### 3.1. AFM1 in Raw Milk in Northern Italy

Monitoring AF contamination in milk is critical as AFM1 causes carcinogenicity, mutagenesis, teratogenesis, genotoxicity, and immunosuppression [33,34,35]. Therefore, since 2004, the Regional Breeders Association of Lombardy has been conducting a survey program to monitor AFM1 contamination in whole raw milk. In this study, the presence of AFM1 was analysed in 95,882 milk samples, both bulk tank milk as well as milk samples collected from farmers, during an eight-year period (September 2013 to July 2021). The results based on mean values of AFM1 contamination indicate that almost all the matrices analysed in this study complied with the legal limits set by the European Union for AFM1 in milk (Table 1). The results showed that only 667 milk samples showed a content of AFM1 exceeding the EU level of 50 ng/L (0.7% of the total number of samples analysed). Positive samples that did not comply with EU limits, consequently, cannot be sold on the market, thus leading to significant economic losses for producers.

Aflatoxins (AFs) are found in a wide range of animal feeds, such as cereal grains, pulses, nuts, dried fruits, etc. [36]. Extensive contamination of animal feed with AFs poses a serious risk also for the dairy industry. In fact, although ruminants are generally more resistant to toxic effects than monogastric animals [37], the main concern is carryover of AFB1 in the form of AFM1 in milk and dairy products [38].

When ruminants eat feed contaminated with AFB1, it is metabolised in several metabolites, mainly AFM1 and AFM2. When eaten, the parent AFs are absorbed and transported by the blood stream into the liver, where bio-activation occurs, resulting in formation of a reactive epoxide at the 8,9-position of the terminal ring [32]. Liver metabolism of AFs can result in production of M1 and M2 metabolites, which can be excreted into urine as well as milk and milk products [39,40,41,42]. Several studies have reported that, depending on the level of feed contamination, approximately 0.3 to 6.3% of AFB1 ingested by livestock is transformed into AFM1 in milk [6,42]. However, it is important to highlight that amount of converted AFB1 into AFM1 is influenced by breed, health, type of diet, milk production, rate of ingestion and digestion, etc. [43]. In particular, carryover in dairy cows milked two times daily was usually 1–2% of the ingested AFB1 for low-yielding cows (<30 L milk yield/day) and up to 6% for high-yielding cows (>30 L milk yield/day), probably due to consumption of significantly higher amounts of concentrated feeds compared to low-yielding cows [44]. Since high-yielding breeds are the prevalent dairy cows farmed nowadays [45], and, due to the increasing extreme hot and droughty season, especially in the south and southwestern regions of Europe (such as Italy), monitoring of AFM1 contamination in raw milk over time is extremely important.

### 3.2. AFM1 in Milk in Northern Italy—Year-on-Year Variation

Looking at mean contamination over the years (Figure 2), AFM1 contamination showed a substantial increase in 2013 and 2015. In fact, while the mean contamination in other years was always below 10 ng/L, the mean AFM1 contamination rose to 14.4 ng/L in 2013 and to 15.1 ng/L in 2015.

Fluctuations in AFM1 levels in milk are due to variations in AFB1 contamination levels in different crops. In 2013, for instance, a major crisis in terms of AFM1 contamination in milk hit Serbia because many raw milk and dairy product samples exceeded the EU maximum residue limits (MRLs) of AFM1 [39,40]. High content of AFM1 in milk and dairy products was reported [41,42,43], probably due to extreme weather conditions in 2012 that increased the AFB1 contamination in animal feeds used for feeding lactating animals. High percentages of maize contaminated by AFs were also reported from several countries in the south and southwestern regions of Europe, such as Spain, Italy, Serbia, Croatia [44,45,46,47,48], as well as Turkey [49], and Middle Eastern countries, such as Iran, Syria, and Egypt [43,44,45].

In Italy, maize is widely grown in the northern regions, and the main concern was contamination with fumonisins, with a high incidence rate in most years. Further, 2003 was the first-time that significant problems arose due to AF contamination of maize [46]. The summer was particularly dry and hot, with maize crops stressed by the lack of water, and, consequently, maize grain was highly contaminated, resulting in problems with AFB1 contamination in feedingstuffs. High levels of AFM1 in milk and derived products were found [46]. Under similar climate conditions, AF contamination of maize also occurred in 2012 in Po Valley, one of the highest-risk areas in Italy in this regard.

With almost 30% of dairy cows of the national heritage and 43.6% of milk production, Lombardy is particularly sensitive to the problem of mycotoxin contamination. Further, 2003 was the first time that Italy was faced with an outbreak related to the effects of climate change on cultivated crops. The summer was particularly dry and hot, with maize crops stressed by drought and lack of water, and the consequence was high AFB1 contamination [15]. Important repercussions have occurred in the dairy industry, with high levels of AFM1 in milk. A similar situation occurred in Italy during the summer of 2015 and lasted until 2016, with an increase in the mean levels of AFM1 in milk (Figure 1) [15]. In our previous study, we analysed contamination of several feed ingredients available in Northern Italy and assessed that maize and cotton are the matrices that show the highest contamination [15]. In particular, both maize and cotton showed peaks in contamination in 2015, with contamination values of 15.7 and 14.1 µg/kg, respectively [15]. Attending to presence of AFs in feed is very important because of possible contamination of milk produced by animals fed with high-AF-contaminated feed. Plotting the mean contamination of raw cow’s milk over the years with the mean contamination of feed ingredients, analysed in the same period by the same laboratory, we noticed interesting pattern similarities. In particular, the results indicated some similarities in the pattern of AFs in both feed and milk. Indeed, the increase in contamination for the year 2015 for AFM1 in milk showed a parallel pattern with the AFB1 contamination both for cotton and for the different maize matrices (Figure 1).

Maize is one of the most important feed ingredients among cereal crops, and safety of its consumption is threatened due to AF contamination [47]. In Italy, maize is widely cultivated in Lombardy, about 31,400 hectares, and, in Po Valley, it is mainly used to produce maize silage for dairy cows. The maize grown in Po Valley is cultivated in an environment with high air humidity and high temperatures, which are extremely favourable for development of the main toxigenic moulds [48]. Ensiling practices, thus, could be positive in terms of mycotoxin control given the data of our previous study, in which maize silage and high-moisture maize were shown to be less susceptible to AF contamination [15]. On the other hand, raw cotton seeds can be a good source of protein for animals, for ruminants in general and dairy cattle in particular, since cotton seeds have some potential in milk production. Thus, their use must be limited due to their heavy AF contamination.

However, the data here reported should be considered with caution as the samples analysed, both feed and milk samples, were collected and analysed under the self-control plan of the Italian dairy industry. For this reason, it was not possible to directly correlate occurrence of AFB1 in dairy cow feedstuffs and AFM1 in the corresponding milk. Although the feed ingredients and milk come from the same geographical area, we cannot conclude that these were consumed by the cows that produced the milk analysed but may have been sold elsewhere. The results indicated that, in general, there are some similarities in pattern of AFs in some feed and milk. Use of these matrices, normally used in high-yielding dairy cow feeding, could increase risk of high concentrations of AFM1 in milk, especially in a climate change scenario that can transform the areas of incidence of AFs. However, further studies on incidence of AFs due to climate change, as well as AF transfer rate from feed to milk based on level of contamination, type of feeding, duration of exposure, type of breeding, etc., are needed.

Given the weather conditions in the second half of 2015, in March 2016, the regional authorities in Lombardy activated the “Extraordinary operating procedures for the prevention and management of the risk of contamination by AFs in the dairy supply chain and in the production of maize”. In 2016, thanks to the control system, presence of some toxins was discovered in certain batches of milk produced in Lombardy and Veneto and, in general, throughout Po Valley. In fact, the summer of 2015 was a very hot season, with high drought peaks, which led to intensified presence of AFs in feed. The type of feed was maize and had been contaminated due to water stress due to the strong heat and drought season. The maximum limit for AFM1 in milk (set by EU legislation) is 50 ng/L, above which human consumption or marketing of milk are not possible. For this reason, from 2016, regional authorities in Lombardy set a level of 40 ng/L in order to allow effective interventions before milk poses a risk to the health of the consumer [49]. When the AFM1 concentration of the analysed milk reaches the action level of 40 ng/L, dairy farms must be informed to apply corrective measures at farm level to avoid high contamination of milk. Among the interventions to be implemented at the farm level, training and awareness were provided for correct supply, storage, and use of animal feed and related raw materials. If corrective measures are not taken at farm level, with confirmation that the concentration of AFM1 has exceeded the legal limit, the plants cannot process the milk with AFM1 content higher than 50 ng/L and the competent authority must be informed in accordance with Italian law [43]. In this study, mean AFM1 contamination from 2016 onwards was lower than in 2015, and percentage of positive samples was relatively constant (Figure 2).

The lower level of AFM1 contamination, as well as the constant and very low percentage of positive samples out of the total number of samples analysed, underline how monitoring programs are very effective in terms of control. Although Mediterranean and Middle Eastern countries have been faced in recent years with climate change, which increases extreme events (very high temperatures and exceptional amounts of rain) that favour AF contamination, the AFM1 concentration in milk and dairy products within the EU is usually at a very low level and well below the MRL [42], thus indicating that strict controls and continuous monitoring of feedstuffs are effective.

Monitoring, verification, and corrective systems have evolved considerably both in terms of prevention (crops, storage of raw materials, “cleaning” treatments in the feed mill) and zootechnical and milk food and derivative checks. The uneven distribution of AFB1 in consignments of zootechnical feed (raw materials, flours, silage) means, however, that, often, the samples taken of feed are not very representative and extremely variable over time, especially in the case of large batches of a particular product, making milk monitoring programs even more effective and useful.

## 4. Conclusions

This study examined a significant volume of data concerning AFM1 contamination of raw milk samples produced in Northern Italy. The AFM1 content in most samples followed EU regulatory levels of 50 ng/L. Additionally, the decline in positivity in recent years was linked to the creation of the Regional Surveillance Plan for AFs, which highlighted the crisis and offered solutions. The study further highlights the connection between feed and milk contamination. Some feedingstuffs seem to be more effective as markers/sentinels of potential carryover between feed and milk, where incidence of AFs is maize > cotton > high-moisture maize > silage maize. Extensive use of these matrices, combined with a change in traditional areas of incidence of aflatoxins risk, could increase risk of increased carryover between feed and milk in the future. However, these results should be considered with caution since, in this study, it was not possible to match the results of contamination in feed with milk because all the samples were collected in the frame of regular quality controls. Chain monitoring, starting with animal feed, is essential to ensure safe production and consumer protection. Activation of the special plan has reduced risks to consumers, keeping the mean AFM1 contamination at very low levels (under 10 ng/L).

Co-occurrence of different AFs and climate change affect the future prospects of predicting the effects of climate change on production and presence of AFs. Monitoring suppliers, purchase of fewer contaminated raw materials, as well as analysis of each batch of animal feed and raw milk help to reduce spread of AF.

## 5. Materials and Methods

### 5.1. Milk Samples

A total number of 95,882 samples of whole raw milk were collected in Northern Italy between September 2013 and July 2021, which were then evaluated for presence of AFM1. Milk samples were collected by dairy farms throughout the territory of Po Valley according to an established sampling protocol and monitoring system decided by the competent regional authorities. After the collection, the samples were transported to the laboratory in refrigerated boxes and stored at +4 °C until analysis. All the samples were collected in the frame of regular quality controls over the monitored period. Due to rights issues, the authors agreed not to identify the sample sources.

### 5.2. Sample Preparation

Refrigerated raw milk samples were centrifuged at 3500× *g* for 10 min at +4 °C. After centrifugation, the upper cream layer was aspirated with a Pasteur pipette, and 100 μL of the skimmed milk was applied directly in the ELISA Bio-Shield M1 ES (purchased from ProGnosis-Biotech, Larissa, Greece) in accordance with the analytical procedure.

### 5.3. Enzyme-Linked Immunosorbent Assay Procedure

Quantitative analysis of AFM1 in samples was performed according to the manufacturer’s instructions. The commercial test kit included standard solutions of 0, 5, 10, 20, 40, and 80 ng/L, which were used to generate regression curves between AFM1 concentration and optical density.

A volume of 100 μL of the AFM1 standard solutions or samples (100 μL/well) was added in duplicate to the wells and incubated for 45 min at room temperature in the dark. The wells were washed three times with 250 μL washing buffer. After washing, 100 μL of the peroxidase-conjugated AFM1 was added and incubated for 15 min at room temperature in the dark. After incubation, the wells were washed again three times with 250 μL washing buffer. Next, 100 μL of substrate/chromogen was added to each well, gently shaking the plate, and the samples were then incubated for 15 min at room temperature in the dark. At the end of incubation, 100 μL of the stop solution was added to each well and mixed gently by shaking the plate manually. Absorbance was measured at 450 nm using the ELISA microplate reader (purchased from Tecan, Switzerland).

Repeatability and reproducibility were analysed using positive controls, which consisted of positive samples of milk contaminated at 30 ng/L AFM1.

Positive samples above the 50 ng/L limit were further confirmed using an immunoaffinity column for clean-up and HPLC-FLD.

### 5.4. Evaluation of AFM1 Concentration

The absorbance values obtained for the standards and samples were divided by the absorbance value of the first standard (zero standards) and multiplied by 100 (percentage maximum absorbance). The absorption was inversely proportional to the AFM1 concentration in the sample. The detection limit of the method was 5 ng/L, whereas samples above the limit of 80 ng/L were diluted and then retested. Milk samples were divided into four categories depending on level of AFM1 contamination: very low risk (<25 ng/L), low risk (25–40 ng/L), moderate risk (40–50 ng/L), and high risk (>50 ng/L-above the legal accepted limit-suspension of sales). This classification is in line with that used by the lab in its reporting to the farmers.

AF concentrations recorded in the same area, in the same laboratory, from the same farms, and period, reported by Ferrari et al. [11], have been analysed in order to evidence similarity and or discrepancies in pattern of milk and feed contamination. Details of determination of AFB1 in animal feed ingredients are provided in a previous publication [11].

### 5.5. Statistical Analysis

The results were analysed using MS Excel. Data were expressed as average content, detection rate, and percentage of samples that exceeded regulatory limits.

## Figures and Tables

**Figure 1 toxins-15-00168-f001:**
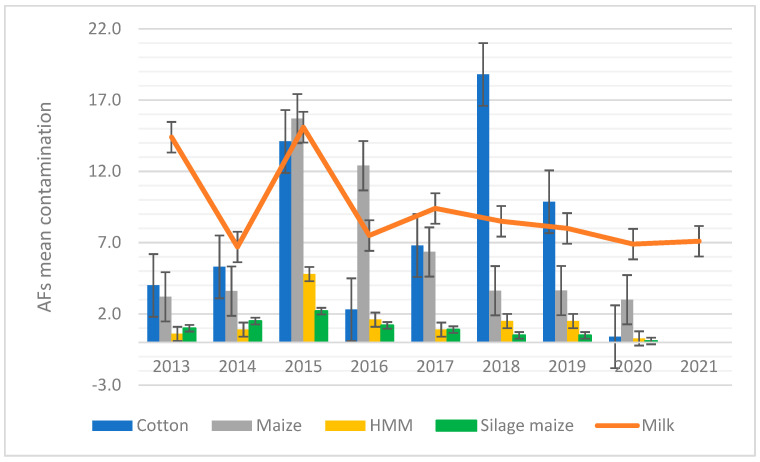
Aflatoxin contamination in milk and different feed ingredients. Year-on-year variation in AFM1 mean contamination in milk (orange) expressed as ng/L and AFB1 mean contamination in cotton (blue), maize (grey), high-moisture maize—HMM (yellow), and silage maize (green) expressed as µg/kg.

**Figure 2 toxins-15-00168-f002:**
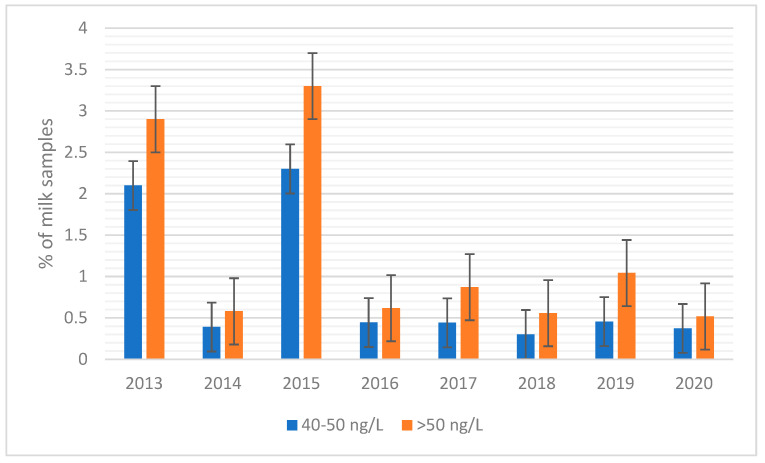
Aflatoxin M1 (AFM1) contamination in milk. Percentage of milk samples under alert (40–50 ng/L AFM1, blue) and percentage of milk samples with AFM1 exceeding the EU threshold of 50 ng/L (orange). The percentage is expressed as the ratio of total samples analysed.

**Table 1 toxins-15-00168-t001:** The levels of AFM1 contamination in raw milk samples analysed.

AFM1 Concentration (ng/L)				
	Very Low Risk (<25)	Low Risk (25–40)	Moderate Risk (40–50)	High Risk (>50)
Number of samples	92,823	2002	390	667
Percentage of samples (%)	96.8	2.1	0.4	0.7

## Data Availability

Data not available to be shared.

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
