# Peer review of "Compliance between Food and Feed Safety: Eight-Year Survey (2013–2021) of Aflatoxin M1 in Raw Milk and Aflatoxin B1 in Feed in Northern Italy"

_toxins, 2023, doi:10.3390/toxins15030168_

Round 1
Reviewer 1 Report
The authors present a manuscript that looks at Aflatroxin levels over an eight year period within the region of Northern Italy
I have a few points
I'm not sure I completely understand the title, especially the part referring to "One Health Perspective". Could the title focus on the main findings/aim of the work in terms of detecting Aflatoxins
I have a similar issue in the abstract, the measurements are made over an 8 year period starting from 2013. But then the authors write in the abstract "The Regional Surveillance Plan (2016) for AFs provided the opportunity to manage and monitor possible excess values." So I'm not certain what point the authors are making here?
Also the next sentence " During the period considered, this incentivised farmers to aim for lower general contamination levels." Considering the proceeding sentence the time period should be defined. But more importantly how do we know farmers were incentivised to aim for lower levels? We need to understand the scenario before and after this monitoring - what has changed. How has this change incentivised farmers? Maybe this is the story that the paper will tell, but it is not obvious from the abstract what the situation was prior to the monitoring, and when the monitoring started. Could this be made more clear.
Also the abstract doesn't mention much about the link to animal feed which is stated as an aim of the study at the end of the introduction "Furthermore, we wanted to determine whether the differences in AFM1 levels in milk were in compliance with the presence of AFB1 in feed ingredients present in the same geographical area."
In Figure 1 why have ytou not drawn a linear line of best fit through the points and given the R2 value rather than stating it is "As can be seen, the calibration 87 curve is virtually linear in the range 0–80ng/L AFM1" Also the graph only displays the concentration range 5-50ng/L so how can we judge this statement from the figure?
Where do the group definitions come from - : i) maximum safety (<25ng/L), ii) safety (25-40ng/L), iii) alert (40-50ng/L), iv) suspension of sales (>50ng/L). Or is this a classification that the authors have created. I'm just not 100% sure about the naming of the categories i.e. maximum safety etc.
In Figure 2 does the mean concentration show anything - would it be better to show a box and whiskers plot or similar that identifies the mean and range etc. for each year
Figure 3,4,5,6 could be combined into one figure with compound bars for the different feedstuffs?
Previously you called 40-50 range alert but in Figure 8 legend you refer to the same range as under alert " Percentage of milk samples under alert (40-50ng/L AFM1, blue) and percentage of milk samples AFM1 exceeding the 140 EU threshold of 50ng/L (orange)"
Actually the % under alert would be much higher?
Can the authors clarify this point "Focusing on levels of AFM1 in milk from 2016 onwards, the percentage of positive samples is important because the mean AFM1 contamination only gives an overview. Figure 7 shows the percentage of positive samples and samples under alert (between 40- 50ng/L)."
I'm not sure it is clear what the findings are and the reason for changing the date range and way of displaying the data etc.
I think if the authors can rectify some of these issues then the paper could be considered for publication
Reviewer 2 Report
The current study "Milk Safety in a One Health Perspective: an eight-year survey (2013-2021) of aflatoxin M1 in raw milk in northern Italy" highlight the overall contamination of raw milk with Aflatoxin M1 in northern Italy. The study have conducted on a fairly large number of samples and results can be significant for the local market but it must be improved in light of following comments.
1. What is the rationale for the study? was there a specific research question other than just a report on contamination level?
2) What was the rationale behind sample collection? Inclusion, exclusion criteria? is it a retrospective study? study location? is it multi-center study? these info are missing in methodology section.
3) Maps showing positivity of samples of different location or a heat map is better presentation
4) any link of aflatoxin positivity with specific breed of animal or managment system?
5) risk factors analysis? can it be done from the existing data?
6) sampling strategy: was it random or convenent sampling? what about the biasness in data because of sampling?
7) Can conclusion drawn be generalized?
8) what about comparison with authors surveys in the region?
Unless the article generate a genuine research question and give appropriate design along with additional data, article can not be considered for publication.
Reviewer 3 Report
Toxins Report: Article # 2168094
Title: Milk Safety in a One Health Perspective: an eight-year survey (2013-2021) of aflatoxin M1 in raw milk in northern Italy.
General comments: The current manuscript summarized 8 years of data (2013-2021) concerning aflatoxin M1 concentrations in raw milk and feed materials in the Lombardy area of Northern Italy. It also evaluated a monitoring system (starting in 2016…) for the potential of lowering rising aflatoxin M1 values due to changing weather patterns, to acceptable levels. The study is relevant in view of the persistent weather changes affecting diet staples of children, in particular.
There are couple problems with the data. The authors highlight the fact that 2012 was a bad year for AF contamination in all southern Europe, yet their aflatoxin data does not start till 2013. This is unfortunate because general contamination of feed grain/silage in 2012 could explain area wide contamination of milk the following year, which is indicated by the sharp peak in 2013. Additionally, because the feed materials and milk were not necessarily obtained from the same area, it cannot be determined where the feed came from that resulted in higher AFM1 values in the milk in certain years. Consequently correlations were not possible and conclusions were based on generalities at best. In spite of these drawbacks, the manuscript presents a need for monitoring milk for toxic contaminants in areas affected by changing weather patterns. A monitoring system put in place in 2016 seems to be effective in decreasing the occurrence of AFM1 in milk, and the feed data also seems to better correlate with the milk data after 2016. Still, Europe has pretty strict regulations and it is not clear if 1% rejection rate is acceptable or not.
Overall the manuscript is relatively well written, needing minor grammatical/stylistic changes suggested below. Also the graphs would benefit with the addition of error bars, indicating statistical significance/relevance.
Specific suggestions:
Line 37: should read…aflatoxin M1…
Line 38: should read…lower than that of the parent…
Line 39: should read…human hepatocarcinogen…
Lines 47-48: sentence rephrased…Strict legal limits, such as those imposed by the EU, cause M1 to have adverse effects on the economy.
Line 55: should read…literature concerning AFM1…
Line 72: should read…than the HPLC…
Line 74: should read…dangerous, recurring…
Line 76-77: should read…decade, Italy and other EU countries also have…
Line 79: should read…AFM1 occurrence in milk.
Line 81: should read…in the Lombardy…
Line 101: should read…samples not surpassing the EU…
Line 128: should read…(AFM1) were plotted against AFB1 in concentrate…*What concentrate?? Please clarify!*
Line 130: should read…and milk, with some differences.
Line 131: should read…contamination shows a…
Line 133: should read…Of note is the situation…
Line 134: should read…in conjunction with a…
Line 136-137: rephrased…On the other hand, silage maize was less contaminated in comparison…
Line149: contamination under control…*What is your definition of control? Clarify please!*
Line 174: should read…of AFB1 in the form of AFM1 in milk…
Line 177: should read…occurs, resulting in…
Line 190: should read…the monitoring of AFM1…
Line 203-204: should read…from several countries…Italy, Serbia and Croatia…
Line 205: should read…, and Middle East countries such as…
Line 207-208: should read…main concern was contamination with fumonisins,
Line 217: should read…In 2003 was the first time…
Line 228-229: should read…we noticed interesting pattern similarities.
Line 290: should read…This study examined…data concerning AFM1…
Line 292: should read…Additionally, the decline…
Line 294-295: should read… The study further highlights the connection between…
Line 300: should read…these results should…
Line 301: should read…contamination in feed with milk…
Line 304: should read…has effectively reduced…*You may want to include % reduction or some other statistical measure.*
Line 307: should read…analysis…
Line 308: should read…milk, help to …
Line 323: A sufficient number?? You may want to explain more precisely what that means!
Line 336: should read…the ELISA microplate…
Reviewer 4 Report
Dear Authors,
The manuscript “Milk Safety in a One Health Perspective: an eight-year survey (2013-2021) of aflatoxin M1 in raw milk in northern Italy” presents relevant information and can be considered for publication in Toxins journal.
The topic is very significant and relevant.
The manuscript is clear and well written.
Minor comments:
· Please leave a space between the value and the unit throughout the manuscript. E.g. “50 ng/L”.
· Line 347: The 5.5. subchapter serial number appears twice.
· Line 347-355: Is a separate subsection really necessary because of one sentence? If yes, I recommend the modification of the subchapter title.
· Several parts (e.g. Author Contributions, Conflicts of Interest, etc.) required by formal requirements are missing from the manuscript.
· Authors need to modify some references according to the requirement of the “Instructions for authors”.
Reviewer 5 Report
The study provides a descriptive overview of AFM1 concentration in milk samples in northern Italy, and attempts to relate those data to contamination of animal feed with AFB1. The paper is interesting and of significance (especially because of the impressive number of samples analyzed and the long timeframe). However, I feel that the study is limited in its academic rigor; only summary statistics are presented, with no statistical tests to reinforce the readers’ confidence in the trends reported. There is some improvement to be made in clarifying the relationships between feed AFB1 and milk AFM1 using the data reported, which would make the paper much stronger. Overall a nice effort and a worthy contribution to the mycotoxin literature.
L3: “above all” is a bit ambiguous. Can you clarify what you mean? Public health importance? Toxicity? Biosynthetic efficiency?
L27: a citation is required here for the sentence beginning “Mycotoxins are produced following…”
L38: citation is required
L59-L72: This paragraph about the merits of various analytical methods seems to verbose given the nature of this study. I would recommend shortening this paragraph and combining it with the one above, to keep the introduction as concise as possible.
L73: This paragraph has some redundancies with the second paragraph in the section; check to ensure that the conceptual relationship between feed AFB1 contamination and AFM1 concentration is clear, and cite relevant sources.
L80: before this concluding paragraph, it would be good to include a paragraph about why milk is important. In the discussion, the nutritive value of milk is alluded to, but very little about milk per se is written in the Introduction. This will make the nutrition content in the Discussion seem more relevant to the article.
L87: The first sentences of the Results should be informative, and should highlight a key finding. For the lay reader, it would be helpful to briefly explain the significance of the standard curve and what its linearity signifies (note that the “results” come before the “methods” in this Journal, so it is important in the Results to narratively articulate why the findings contribute significantly to the study’s objectives)
L90: in the figure legend, specify what type of assay this calibration curve is derived from
L92: How were these categories (and their concentration cutoffs) determined? Explain in the methods why these cutoffs are significant. Regarding the use of “Maximum Safety” and “Safety” – can any amount of contamination with this toxin be considered “safe”? Perhaps rephrase these category names in terms of risk, i.e. “Very Low Risk,” “Low Risk,” “Moderate Risk,” “High Risk.”
L95: for clarity, I have the following recommendations for formatting Table 1: 1) capitalize first letters in the category headings, 2) put parentheses () around the category cutoff concentrations, change the word “Distribution” to “Percent of samples”
L100-101: the part about “farm level” intervention seems like it belongs in the discussion, not in Results. Also: I don’t understand why this finding indicates that farm-level corrective action is needed. Please clarify this aspect in the discussion
L106: were these differences statistically significant? It would be appreciated if results of a statistical test were presented in the results (and corresponding methods described in the Methods section)
L107-109: again, this part belongs in the Discussion
Figures 4-6 could ideally be combined into a single figure, either overlaid or in multiple panels. It is not necessary to show 3 different figures, as these plots communicate similar ideas and trends. At the least, kindly ensure that the y-axis ranges are the same across all three plots.
L143: why was this part of the analysis specifically for 2016 onward, and not the full range? Please clarify here and in the methods.
L152: this is the first mention of nutrition in the paper, and it seems a bit out of place. See my comment above regarding introducing this concept first in the Introduction. Also: please cite a source of evidence regarding the nutritional value of milk for infants/children.
L175: This sentence is not clear. What does “originate” mean in this context, and what does “taken from” mean? Please use more precise language to communicate these ideas. “most important” is superlative language, which should be avoided. Use language like “among the most important,” or simply “important”
L218: What was the nature of the mentioned outbreak? Can a source be cited?
L251: It would be good to cite some evidence here that demonstrates the importance of this limitation, i.e. that there can be marked variability in contamination within and across batches of feed/milk
L259: What “crisis”? This doesn’t seem to transition from the paragraph above. This is the first time that the situation is portrayed as a “crisis,” which is rather extreme language. Perhaps use more neutral language like “due to the concerning levels of AFM1 identified in…”
L288: Would be good to also mention the regulatory landscape. Are AF levels in feeds regulated in Italy? Would that be an example of farm-level corrective action, like what was mentioned in the Results?
L312: were the samples collected at the same intervals/seasons in each year? Please specify how many sample collection events there were per year, and whether there was consistency across years in seasonal distribution of sampling effort.
L347: Methods sub-section 5.5 needs to have some additional detail. I believe this is the sub-section that elaborates on the analysis that produced Figures 4-6. If so, please give more details about the source(s) of the feed contamination data and clarify the relationship between those values and the AFM1 estimates.
L348: “AFL” is used here, whereas “AF” is used in other sections. Please be consistent with abbreviations
Round 2
Reviewer 1 Report
The authors have addressed my comments and made extensive changes to the figurtes and text within the manuscript and I feel that the manuscript could be published in toxins
Reviewer 2 Report
Manuscript has been improved with a better presentation of results.